# Influence of Temperature Reaction for the CdSe–TiO_2_ Nanotube Thin Film Formation via Chemical Bath Deposition in Improving the Photoelectrochemical Activity

**DOI:** 10.3390/ma13112533

**Published:** 2020-06-03

**Authors:** Chin Wei Lai, Nurul Asma Samsudin, Foo Wah Low, Nur Azimah Abd Samad, Kung Shiuh Lau, Pui May Chou, Sieh Kiong Tiong, Nowshad Amin

**Affiliations:** 1Level 3, Block A, Nanotechnology & Catalysis Research Centre (NANOCAT), Institute for Advanced Studies (IAS), University of Malaya, Kuala Lumpur 50603, Malaysia; nurazimah.sam@gmail.com (N.A.A.S.); lauks_1010@yahoo.com (K.S.L.); 2Institute of Sustainable Energy (ISE), Universiti Tenaga Nasional (The Energy University), Jalan IKRAM-UNITEN, Kajang 43000, Selangor, Malaysia; Siehkiong@uniten.edu.my (S.K.T.); Nowshad@uniten.edu.my (N.A.); 3School of Engineering, Faculty of Built Environment Engineering, Technology & Design, Taylor’s Lakeside Campus, No. 1, Jalan Taylors, Subang Jaya 47500, Malaysia; puimay.chou@taylors.edu.my; 4Department of Electrical, Electronic and Systems Engineering, Faculty of Engineering and Built Environment, Universiti Kebangsaan Malaysia (National University of Malaysia), Bangi 43600, Malaysia

**Keywords:** electrochemical anodization, chemical bath deposition, photoelectrochemical activity, cadmium selenide, CdSe–TiO_2_ nanotube thin films

## Abstract

In this present work, we report the deposition of cadmium selenide (CdSe) particles on titanium dioxide (TiO_2_) nanotube thin films, using the chemical bath deposition (CBD) method at low deposition temperatures ranging from 20 to 60 °C. The deposition temperature had an influence on the overall CdSe–TiO_2_ nanotube thin film morphologies, chemical composition, phase transition, and optical properties, which, in turn, influenced the photoelectrochemical performance of the samples that were investigated. All samples showed the presence of CdSe particles in the TiO_2_ nanotube thin film lattice structures with the cubic phase CdSe compound. The amount of CdSe loading on the TiO_2_ nanotube thin films were increased and tended to form agglomerates as a function of deposition temperature. Interestingly, a significant enhancement in photocurrent density was observed for the CdSe–TiO_2_ nanotube thin films deposited at 20 °C with a photocurrent density of 1.70 mA cm^−2^, which was 17% higher than the bare TiO_2_ nanotube thin films. This sample showed a clear surface morphology without any clogged nanotubes, leading to better ion diffusion, and, thus, an enhanced photocurrent density. Despite having the least CdSe loading on the TiO_2_ nanotube thin films, the CdSe–TiO_2_ nanotube thin films deposited at 20 °C showed the highest photocurrent density, which confirmed that a small amount of CdSe is enough to enhance the photoelectrochemical performance of the sample.

## 1. Introduction

Nanostructured titanium dioxide (TiO_2_) has been known as one of the most promising semiconductor materials, as it has been widely used in many applications, such as photocatalysts [1,2,3,4], photovoltaics [5,6,7,8,9,10,11], photoelectrochemical cells [12,13,14,15], supercapacitors [16,17,18,19], and sensors [20], due to its remarkable chemical, optical, and physical properties, as well as its low production cost and lower toxicity. The electrochemical anodization method has come to light, as it is proven to be the most facile and versatile method to synthesized TiO_2_ nanotube thin films due to its ability to modify the morphology, diameter, and length of the nanotubes by varying the anodization parameters. In addition, their unique nanoarchitecture minimizes the photo-induced charge of the recombination loss of the carrier at the nanostructure connections, thus, maximizing photon absorption [21,22]. However, due to its wide band gap (3.2 eV for the anatase phase and 3.0 for the rutile phase TiO_2_), the adsorption by the material is limited to UV wavelength [23]. Moreover, the fast electron/hole recombination and sluggish charge transfer by the TiO_2_ nanotube thin films significantly demerit the photoelectrochemical performance of this material [24]. In order to overcome the limitations, modification, such as electrochemical reduction [25,26], metal oxide heterojunction [27,28], defect engineering [29], and metal or non-metal doping [30,31,32,33], have been done to lower the band gap of the TiO_2_ nanotube thin films and, thus, enhance the conductivity, radiation adsorption, and catalytic activity, respectively. The deposition of small band gap semiconductor materials as a sensitizer on the TiO_2_ nanotube thin films [34,35,36] have been proven to be an important approach in order to tune the overall band gap of the material and improve its photoelectrochemical performance. By incorporating these materials on the TiO_2_ nanotube thin films, the further enhancement of the adsorption of visible light and fastening the charge transfer of photoexcited electrons into the TiO_2_ conduction band is needed.

Cadmium selenide (CdSe) is one of most remarkable semiconductor materials in Group II–VI, that is being used to sensitize wide band gap TiO_2_ nanotube thin films [23,37]. The CdSe has a lower band edge than TiO_2_, thus, allowing band edge alignment, leading to efficient electron transfer. The band edge alignment effect will improve photo-illumination absorption at visible or near-IR photons and eventually reduce the recombination change carrier losses [38,39,40]. There are several methods that have been used to incorporate CdSe on the nanostructured TiO_2_, which is functional under highly controlled conditions [41,42,43,44]. Among the deposition method, chemical bath deposition (CBD) is one of the methods that has been widely used to deposit CdSe particles on the TiO_2_ nanotube thin films [45]. This method becomes favourable, as it uses simple equipment and apparatus, making this method cost-effective. Our previous report shows that by tuning the parameters of the CBD, the morphology and other physical properties of the CdSe–TiO_2_ nanotube thin films can be altered, thus, enhancing the photoelectrochemical performance of the sample [37,46]. Despite many reports on the aim to increase the performance of CdSe–TiO_2_ nanotube thin films as photoelectrochemical cells, it is crucial to study the deposition parameters, which influence the chemical and physical properties of CdSe–TiO_2_ nanotube thin films. 

The CBD of CdSe used in this work could offer good controls of amount loading, size, and uniformity of CdSe on the TiO_2_ nanotube thin films. Parameters, such as the temperature of the CBD, becomes a crucial role in manipulating the thermal dislocation of the Cd complex and Se anion by controlling the rate of ions released from the precursor, and thereby affecting the growth rate of CdSe on the TiO_2_ nanotubes thin films. With increasing temperature, thermal dissociation of the CdSe precursor is used in this study: sodium selenosulfate (Na_2_SeSO_3_) and cadmium acetate dihydrate (Cd(CH_3_COO)_2_ 2H_2_O) will be increased significantly. Henceforth, increasing the kinetic energy and driving force of the active precursor leads to the increase of interaction between CdSe and nanotubular structures. Last but not least, a comprehensive study has been conducted that aims to optimize the temperature of the CBD of CdSe on the TiO_2_ nanotube thin films in order to enhance their photoelectrochemical performance.

## 2. Experimental Section

### 2.1. Preparation of TiO_2_ Nanotube Thin Films 

The TiO_2_ nanotube thin films were prepared by the electrochemical method of anodizing Ti foils in a two-electrode cell at 40 V for 60 min, consisting of Ti foil as the working electrode and platinum rod as the counter electrode (Figure 1). The electrolyte comprised of ethylene glycol, 0.3 wt % ammonium fluoride, and 5 wt % hydrogen peroxide. The anodized samples were rinsed with deionized (DI) water and subsequently annealed for 4 h in the air at 400 °C with a heating rate of 5 °C/min^−1^, and cooled naturally to achieve the pure anatase phase of TiO_2_ [35]. 

### 2.2. Preparation of CdSe–TiO_2_ Nanotube Thin Films 

The CdSe was deposited on the TiO_2_ nanotube thin films via CBD, as shown in the schematic diagram in Figure 1. To prepare the 5 mM CdSe solution precursor, the selenide precursor (Na_2_SeSO_3_ solution) was prepared by mixing 0.6 M sodium sulfite with 0.2 M selenium metal powder in DI water. The mixture was heated under reflux at 90 °C for 3 h to form a clear Na_2_SeSO_3_ solution. The cadmium precursor was prepared by dissolving 0.2 M of Cd(CH_3_COO)_2_∙2H_2_O in DI water. Then, a concentrated ammonia solution (30%) was added into the cadmium solution slowly to adjust the pH of the solution between 12 to 12.5. This was to prevent a reverse reaction of Cd(NH_3_)_4_^2+^ to form stable cadmium hydroxide (Cd(OH)_2_). Both precursor solutions were then mixed until homogenous. The annealed anatase TiO_2_ nanotube thin films were soaked vertically inclined into the CdSe bath solution for the deposition process for 1 h. The deposition temperature was varied from 20 to 60 °C. A digital programmable hotplate (Torrey Pines Scientific) was used in this experiment. The temperature of the chemical bath was set and, upon the deposition process, the solution’s temperature was checked, using a thermometer for confirmation. Throughout the deposition process, the temperature of the chemical bath was checked from time to time. Then, the samples were the rinsed with DI water and dried in an oven overnight. 

### 2.3. Characterizations

The surface and cross-sectional morphologies of the prepared samples (TiO_2_ nanotube thin films and CdSe/TiO_2_ nanotube thin films) were investigated via field emission scanning electron microscopy (FE-SEM, JEO JSM 7600-F, JOEL Ltd., Tokyo, Japan) while the elemental analysis was conducted using Oxford Instruments by means of energy dispersive X-ray spectroscopy (EDX, Oxford Instruments, Abingdon, United Kingdom) The phase transition and crystallinity of all prepared samples was studied using X-ray diffraction (XRD, Bruker AXS D8 Advance, Bruker, Karlsruhe, Germany)and a Raman spectrometer (Renishaw inVia, Renishaw, Wotton-under-Edge, United Kingdom)In this study, XRD was operated using Cu Kα radiation (λ = 0.1546 nm) at a scanning rate of 2° min^−1^ over the angles 2θ = 20° to 70°. The Raman spectroscopy was operated at an excitation wavelength of 532 nm generated by an Ar ion laser over the range of 100 to 1000 cm^−1^. The binding energy and chemical state of the samples were measured and quantified with X-ray photoelectron spectroscopy (XPS, PHI Quantera II, Physical Electronic, Minnesota, USA) with an Al cathode scan (*hv* = 1486.8 eV) of 100 microns and 280 eV with pass energy. The precise analysis of the features of each element was decomposed in the Shirley background using the Voigt curve fitting function. The optical properties of the prepared samples were recorded using a UV-Vis diffused reflectance spectrophotometer (Shimadzu UV-2700 UV-Vis, Shimadzu, Kyoto, Tokyo) from 240 to 800 nm.

### 2.4. Photoelectrochemical Testing

A three-electrode system composed of Pt as the counter electrode, Ag/AgCl (3 M KCl) as the reference electrode, and the prepared samples as the working electrode were used in a 1 M KOH aqueous electrolyte. The photoelectrochemical measurement of the prepared samples was evaluated by linear sweep voltammetry using a potantiostat (Autolab PGSTAT 204, Metrohm, Herisau, Switzerland) and illuminated by a 150 W Xenon lamp (Zolix LSP-X150, Zolix Instruments CO., Ltd., Beijing, China) with a light intensity of 100 mW cm^−2^ focusing on the dipped part of the electrode.

## 3. Results and Discussion

The surface and cross-section morphologies of all the prepared CdSe–TiO_2_ nanotube thin film samples deposited at various temperatures from 20 to 60 °C are shown in Figure 2a–d. The highly ordered TiO_2_ nanotube thin films were successfully synthesized in our previous report, with an average inner-tube diameter and tube length of 76 nm and 5.6 µm, respectively [21,47]. In this study, a clear surface morphology change was observed for all samples as function to the chemical bath temperature. At a low temperature (20 °C), the deposition of CdSe on the TiO_2_ nanotube thin films was almost negligible, as shown in Figure 2a. However, the average inner-tube’s diameter of the sample decreased to 71 nm upon the deposition of CdSe, indicating that the nucleation process had taken place on the sample. A low chemical bath temperature of 20 °C led to a low deposition rate, thus, slowing the nuclei growth on the TiO_2_ nanotube thin films. As the chemical bath temperature increased to 40 °C, the presence of CdSe particles was observed, with an average inner-tube diameter of 90 nm, which partially covered the nanotubes (refer to Figure 2b). Moreover, the tube’s diameter decreased to 68 nm upon the deposition of CdSe on the TiO_2_ nanotube thin films. The agglomeration of CdSe into larger particles was observed when the chemical bath temperature increased to 50 and 60 °C. They were uniformly dispersed on the surface of the nanotubes with an average diameter of 145 nm at chemical bath temperatures of 50 °C and 220 nm at 60 °C. Furthermore, the agglomeration of CdSe covered most of the nanotube circumferences, as shown in Figure 2c–d. The average tube diameters of the samples were found to decrease to 57 nm (50 °C) and 54 nm (60 °C). The cross-sectional morphologies (inset figures) showed an adhesive texture, with the presence of CdSe particles deposited throughout the wall surface. This indicates that the deposition of CdSe took place on the top surface and the wall surface of the nanotubes. The EDX analysis confirmed the presence of cadmium (Cd), selenium (Se), titanium (Ti), and oxygen (O) elements in all samples, as shown in Table 1. These results supported the observation from the FE-SEM results, as the elemental composition of Cd and Se increased as the chemical bath temperature increased. In addition, the ratio of the Cd and Se elements in all samples was 1:1, thereby confirming the deposition of stoichiometry CdSe on the TiO_2_ nanotube thin films.

Figure 3 shows the XRD patterns for the CdSe–TiO_2_ nanotube thin films deposited at different chemical bath temperatures at 2θ = 20° to 70°. All samples exhibited an intense peak of CdSe at 2θ = 25.35°, which corresponded to the (1 1 1) preferred orientation of a cubic phase of CdSe (JCPDS No.: 19–0191). This peak was found to overlap with the peak of the anatase crystal plane at 2θ = 25.37° (1 0 1) phase. The intensity of the peak increased as function of the chemical bath temperature, which indicated an abundancy of the CdSe and the well crystalline nature of the CdSe [44]. These results were agreeable with the FE-SEM analysis, as the chemical bath temperature increased as the amount of CdSe deposited onto the TiO_2_ nanotube thin films increased. A small peak was observed at 2θ = 49.7°, corresponding to a (3 1 1) plane of cubic zinc blended phase CdSe in the CdSe–TiO_2_ nanotube thin films deposited at the chemical bath temperatures of 50 and 60 °C. However, the peak was not detected in the CdSe–TiO_2_ nanotube thin films deposited at low chemical bath temperatures due to the low content of CdSe (< 2 at % from EDX analysis) in the sample, which was too insufficient to be detected by XRD analysis [48,49,50]. The single phase of TiO_2_ crystal indexed to anatase planes (JCPDS No.: 21–1272) was observed at 2θ = 38.67°, 48.21°, 54.10°, 55.26°, 62.66°, and 68.74°, corresponding respectively to planes (1 1 2), (2 0 0), (1 0 5), (2 1 1), (2 0 4), and (1 1 6). In addition, the peaks at 2θ = 35.1°, 38.4°, 40.2°, and 53.0° corresponded to (1 0 0), (0 0 2), (1 0 1), and (1 0 2), and were originated from a Ti metal substrate (JCPDS no.: 44–1294).

The Raman spectra of the CdSe–TiO_2_ nanotube thin films (shown in Figure 4) further confirmed the presence of cubic CdSe phase in the samples. The characteristic peaks of the anatase phase of TiO_2_ were observed at 144, 394, 515, and 636 cm^−1^, corresponding to E_g_, B_1g_, A_1g_, and E_g_ vibration modes. The E_g_ vibration mode was mainly caused by symmetric stretching vibration, while B_1g_ was due to the symmetric bending vibration, and A_1g_ was caused by the anti-symmetric bending of O–Ti–O in TiO_2_ [46,51]. A gradual decrease in the intensity of these peaks was observed with an increase of the chemical bath temperature due to the increase of cubic CdSe content in the samples. The samples prepared at chemical bath temperatures of 40, 50, and 60 °C showed presence of a Raman shift at 206 cm^−1^, corresponding to the first order for the longitudinal optical phonon (LO) of cubic CdSe.

The surface chemical state of pure TiO_2_ nanotube thin films and CdSe–TiO_2_ nanotube thin films was investigated using XPS analysis. Figure 5 displays the XPS spectra of both samples, which mainly consisted of Ti 2p, O 1s, Cd 3d, and Se 3d elements. Two broad peaks centered at 458.80 and 464.50 eV are observed in Figure 4a, corresponding to the Ti 2p_3/2_ and Ti 2p_1/2_ peaks of Ti^4+^ in the TiO_2_ nanotube thin films [39]. The intensity of both peaks gradually decreased upon the deposition of CdSe into the TiO_2_ nanotube thin films, indicating the presence of CdSe in the TiO_2_ lattice structure. However, the deposition of CdSe on TiO_2_ did not affect the Ti chemical state of the nanotubes. In contrast, the O 1s elements showed a slight shift from 530.90 eV for the pure TiO_2_ nanotube thin film samples to 529.90 eV for the CdSe–TiO_2_ nanotube thin film samples due the strong interaction between CdSe clusters and TiO_2_ nanotube thin films. The strong interaction bond between CdSe and TiO_2_ served as the heterojunctions that became captives of the electron/hole pairs from the CdSe clusters [38]. Hence, this will suppress the recombination of the electron/hole and photoelectrochemical activity.

Figure 5c illustrates the double peak features of the Cd 3d spectra for the CdSe–TiO_2_ nanotube thin film sample, suggesting the presence of Cd element in the sample, while no traces of Cd element were observed in the TiO_2_ nanotube thin film sample. The precise assessment of the functions of the spin-orbital split Cd 3d_5/2_ and Cd 3d_3/2_ were disintegrated by using the Voigt curve fitting function within the Shirley background. Four peaks were observed, centered at 405.13 and 404.59 eV corresponding to Cd 3d_5/2_, whereas the binding energy at 411.98 and 411.40 eV corresponded to Cd 3d_3/2_. The prominent binding energy centered at 405.13 and 411.98 eV, which was due to the core level of Cd^2+^ cations of the CdSe crystal structure [41]. The additional peaks observed with lower intensities of 405.13 and 411.98 had a weak shift in peak positions. CdO formation was also proved in CdSe, where the core levels of the oxidation of Cd were weak at the 3d_5/2_ and 3d_3/2_ core levels [51,52]. The small CdO volume could, therefore, be verifiable using the CBD technique from low intensity peaks with 404.59 and 411.40 bonding energy. The existence of the Se element in the CdSe–TiO_2_ nanotube thin films sample was proved from the Se 3d spectra shown in Figure 5d. A broad peak centered at 54.46 eV represented that the Se 3d_3/2_ characteristic was due to the Se^2−^ anions of CdSe [53,54]. 

The optical properties of pure TiO_2_ nanotube thin films and CdSe–TiO_2_ nanotube thin films were carried out using a UV-Vis diffuse reflectance analysis (DRS). The adsorption spectra for both samples are shown in Figure 6. The adsorption edge of pure TiO_2_ nanotube thin films was observed below 400 nm due to the excitation of the electron from the valence band to the conduction band [54]. However, the CdSe–TiO_2_ nanotube thin films showed an adsorption above 400 nm, suggesting that more visible light was adsorbed by the sample, hence facilitating the generation of a high amount of carrier charge, which then enhanced the photocurrent response of the sample. The Tauc approach was used to determine the optical bandgap of the pure TiO_2_ nanotube thin films and CdSe–TiO_2_ nanotube thin film samples using the equation below:(1)$$\alpha =\frac{\alpha_{0}\sqrt{hv-E_{g}}}{hv}$$
where *E_g_* is the separation between the bottom of the conduction band and the top of the valance band, *hv* is the photon energy, and *n* is the constant. In this case, *n* = ½, which follows a direct transition. The plot of *(αhv)^1/2^* against *hv* for the pure TiO_2_ nanotube thin films and CdSe–TiO_2_ nanotube thin films are shown in Figure 7. The band gap of pure TiO_2_ nanotube thin films and CdSe–TiO_2_ nanotube thin films were estimated from the intersection of the baseline with the tangent line of the sharply decreasing region of the spectra. It was noted that the band gap of the pure TiO_2_ nanotube thin films was 3.20 eV, while the band gap of the CdSe–TiO_2_ nanotube thin films was 2.81 eV. It was well known that the CdSe bulk sample band gap was 1.74 eV. Typically, the band gap of CdSe varies with the deposition method and deposition parameters, and is also attributed due to the changes in the film composition and structural defects. In this case, the large band gap of the CdSe–TiO_2_ nanotube thin films corresponded to the low amount of CdSe on the TiO_2_ nanotube thin films, as discussed in the EDX results. 

A photoelectrochemical analysis of pure TiO_2_ nanotube thin films and CdSe–TiO_2_ nanotube thin films deposited at chemical bath temperatures from 20 to 60 °C was performed using linear sweep potential voltammetry with applied potential from −1 to 1 V versus Ag/AgCl in a 1 M KOH aqueous electrolyte. Figure 8 shows the *J_p_-V* curves for all samples. Upon irradiation, the photocurrent density of CdSe–TiO_2_ nanotube thin films deposited at 20 °C showed the highest photocurrent density value at 1.70 mA cm^−2^, which was 17 % higher than the pure TiO_2_ nanotube thin films (1.45 mA cm^−2^). However, the CdSe–TiO_2_ nanotube thin films deposited at higher chemical bath temperatures exhibited a decrease in photocurrent density, with 1.1 mA cm^−2^, 1.8 mA cm^−2^, and 0.5 mA cm^−2^ for the samples deposited at temperatures of 40, 50, and 60 °C, respectively. The obtained photocurrent density was much higher compared to other TiO_2_ thin film-based photoelectrochemical cells that were reported earlier: Gd@TiO_2_ NRA (~0.51 mA cm^−2^) [30], TiO_2_/g-C_3_N_4_ (~0.85 mA cm^−2^) [55], TiO_2_/WO_3_ (~1.00 mA cm^−2^) [56], and CdSe–TiO_2_ (~1.6 mA cm^−2^) [57].

The obtained results indicated that the low loading of CdSe on the TiO_2_ nanotube thin films was sufficient to enhance the photocurrent density of the sample. Furthermore, the CdSe–TiO_2_ nanotube thin films deposited at a temperature of 20 °C showed a clear surface morphology without any clogged nanotubes. Therefore, the sample experienced better ion diffusion, thus, enhancing the photocurrent density of the sample. Generally, the CdSe species in the sample acted as an effective electron acceptor, which generated an energy band below the conduction band of TiO_2_. Thus, the photogenerated electrons from TiO_2_ will be captured by the CdSe species due to the interpretation of inter-band states with key redox potential, or surface states within the corresponding energetic position. In contrast, in pure TiO_2_ nanotube thin films, the photo-induced electron in the conduction band was trapped for the recombination with holes. In this study, excessive loading of the CdSe species on the TiO_2_ nanotube thin films exhibited poor photocurrent density. As discussed in the FE-SEM results, the CdSe–TiO_2_ nanotube thin films deposited at temperatures of 40 to 60 °C displayed agglomerated CdSe particles, where they mostly covered the TiO_2_ nanotube thin film circumstances and led to poor ion diffusion throughout the nanotubes during the photocurrent analysis. This condition prevented the electron transport throughout the sample and resulted in low photocurrent density [58,59]. Thus, the enhancement of photocurrent density under solar illumination was ascribed to the optimum content of the CdSe species incorporated into the lattice of TiO_2_. 

### Mechanism Study of CdSe–TiO_2_ Nanotubes via CBD Method 

CBD consists of two important steps, which are the nucleation process and particle growth. This method is based on the formation of the solid phase from a precursor solution. During the deposition process, the first stage corresponds to the initiation of the critical nuclei of the species. In this study, a bath solution with a Cd^2+^ and Se^2−^ precursor was used to deposit CdSe on the TiO_2_ nanotube thin films. The mechanisms of the CdSe formation during the CBD have been described in our previous report [37]. Cd(CH_3_COO)_2_·2H_2_O will form a Cd complex with ammonia solution to finally form Cd^2+^.
(2)$$\mathrm{Cd}{\left({\mathrm{CH}}_{3}\mathrm{COO}\right)}_{2}+2{\mathrm{NH}}_{4}\mathrm{OH}\to \mathrm{Cd}{\left(\mathrm{OH}\right)}_{2}+2{\mathrm{NH}}_{4}\left({\mathrm{CH}}_{3}\mathrm{COO}\right)$$
(3)$$\mathrm{Cd}{\left(\mathrm{OH}\right)}_{2}+4{\mathrm{NH}}_{2}\mathrm{OH}↔\mathrm{Cd}{\left({\mathrm{NH}}_{3}\right)}_{4}{}^{2+}+2{\mathrm{OH}}^{+}+4H_{2}O$$
(4)$$\mathrm{Cd}{\left({\mathrm{NH}}_{3}\right)}_{4}{}^{2+}↔{\mathrm{Cd}}^{2+}+4{\mathrm{NH}}_{3}$$

In addition, Na_2_SeSO_3_ will hydrolyze in the solution to produce Se^2-^, as presented below:(5)$${\mathrm{Na}}_{2}{\mathrm{SeSO}}_{3}+{\mathrm{OH}}^{-}\to {\mathrm{Na}}_{2}{\mathrm{SO}}_{4}+{\mathrm{HSe}}^{-}$$
(6)$${\mathrm{HSe}}^{-}+{\mathrm{OH}}^{-}\to {\mathrm{Se}}^{2-}+H_{2}O$$

The Cd(NH_3_)_2_^2+^ then reacts with Se^2-^ to form a CdSe species to be deposited on the TiO_2_ nanotube thin films.
(7)$$\mathrm{Cd}{\left({\mathrm{NH}}_{3}\right)}_{4}{}^{2+}+{\mathrm{Se}}^{2-}\to \mathrm{CdSe}+4{\mathrm{NH}}_{3}$$

The formation of CdSe and the deposition of the species on the TiO_2_ nanotube thin films will occur simultaneously in the chemical bath. The rate of deposition will depend on the deposition parameters, such as chemical bath temperature, pH, precursor’s concentration, and deposition time. The CdSe nuclei will form and initiate the deposition process on the TiO_2_ nanotube thin film surfaces. During this stage, the metastability of the precursor solution system is not affected by the emerging nuclei. As the deposition process progresses, the intermediate stage, which comprises a combination process of growth of the existing CdSe particles, and the initiation of additional nuclei will reduce the metastability of the precursor solution system. These processes will resume until the end of the deposition process.

Figure 9 shows a simple schematic diagram representing the carrier transfer processes between CdSe and the TiO_2_ nanotubes, which can aid the understanding of the PEC mechanism. When the solar light illuminates, both TiO_2_ and CdSe harvest photons and generate electron-hole pairs. The conduction band (CB) of CdSe is more positive than the TiO_2_ nanotubes, therefore, the photo-induced electrons in the CB of CdSe will transfer to the CB of the TiO_2_ nanotubes and further move along the TiO_2_ nanotubes, to the external circuitry [40,59,60]. Simultaneously, the left holes in the valance band (VB) of TiO_2_ will transfer to the VB of CdSe. Therefore, this results in a parallel array of rapid transfer pathways for carrier transportation. The band gap narrowing, which improvises the visible light absorption, will provide more sites that can slow down the recombination of charge carriers [50,51,61]. The holes migrating to the surface of CdSe will react with the electrolyte at the interface between the photoanode and the electrolyte [59]. Based on the aforementioned scenario, CdSe has two advantages: first, CdSe enhances the photon-induced charge separation rate by widening the response spectrum compared with the TiO_2_ nanotubes; second, the heterojunction that anchors CdSe on the surface of the TiO_2_ nanotubes provides additional channels for the separated charge transportation thanks to the specific band potential distribution and the charge transfer property, so that the recombination probability decreases, which results in a higher PEC performance of the sample. In short, the PEC properties of the CdSe–TiO_2_ nanotubes are demonstrated to be controllable through the chemical bath deposition’s temperature.

## 4. Conclusions

In summary, CdSe–TiO_2_ nanotube thin films were successfully synthesized using CBD by controlling the chemical bath temperature. The particle size and CdSe loading on the TiO_2_ nanotube thin films were controlled by varying the chemical bath temperature from 20 to 60 °C. The CdSe–TiO^2^ nanotube thin films showed an excellent photo response in the visible light region, thus, facilitating the generation of a high amount of carrier charge. The highest photocurrent density of 1.70 mA cm^−2^ was obtained by the CdSe–TiO_2_ nanotube thin films deposited at 20 °C, which affirmed that low loading of CdSe on the TiO_2_ nanotube thin films was sufficient to enhance the photocurrent of the sample. This study has brought light to the CdSe–TiO_2_ nanotube thin films as promising materials for photoelectrochemical cell application. 

## Figures and Tables

**Figure 1 materials-13-02533-f001:**
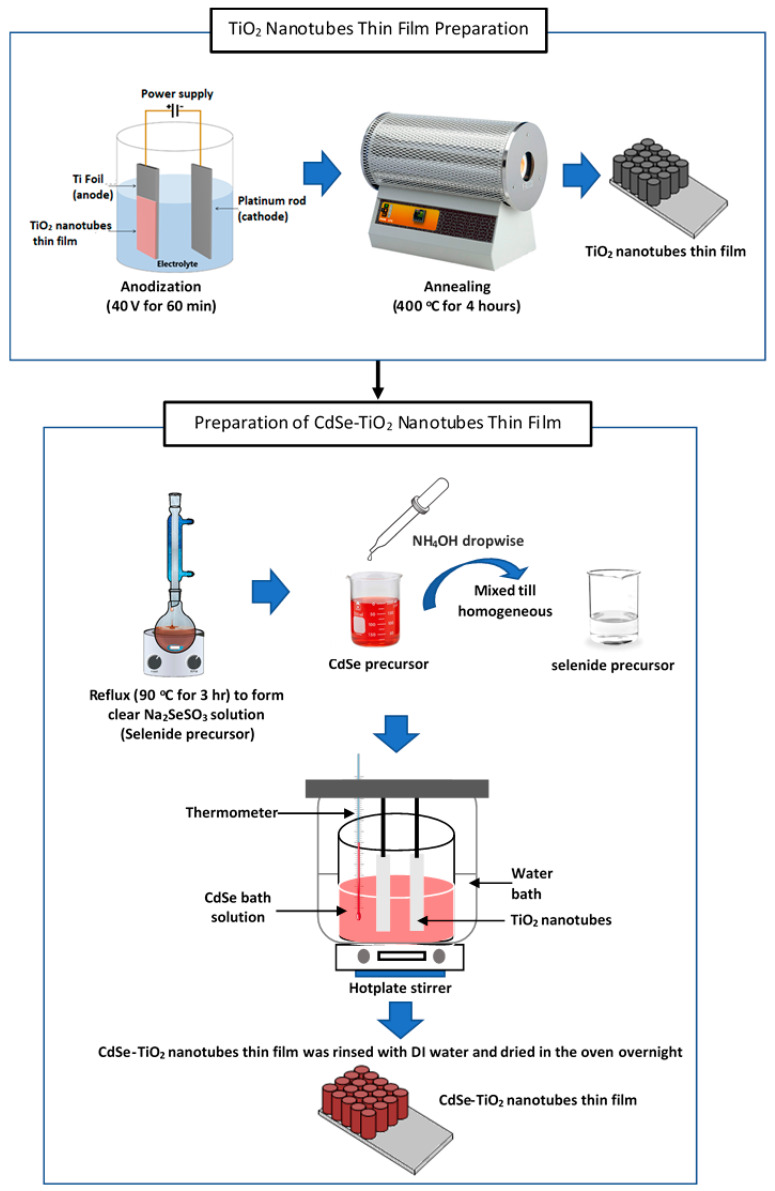
Schematic diagram of the preparation of TiO_2_ nanotube thin films and CdSe–TiO_2_ nanotube thin films.

**Figure 2 materials-13-02533-f002:**
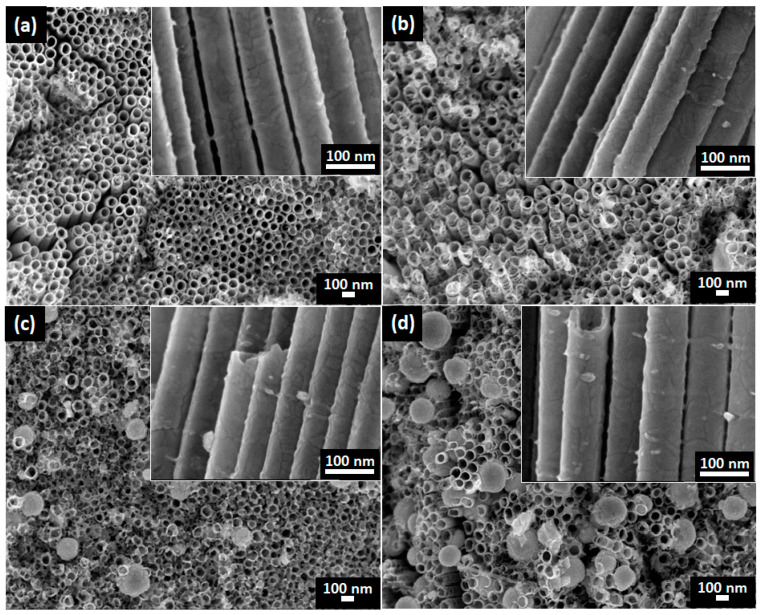
FE-SEM images of CdSe–TiO_2_ nanotube thin films deposited at chemical batch temperatures of: (**a**) 20 °C, (**b**) 40 °C, (**c**) 50 °C, and (**d**) 60 °C, and the inset is cross-sectional view.

**Figure 3 materials-13-02533-f003:**
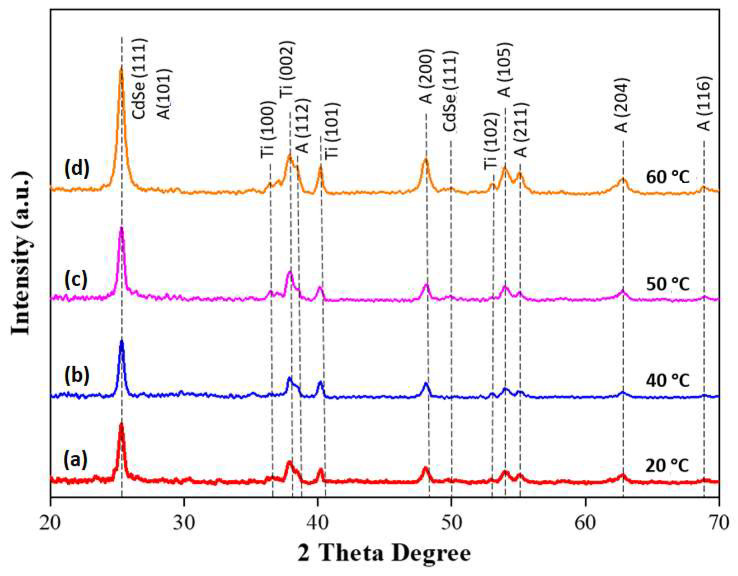
XRD diffraction patterns of CdSe–TiO_2_ nanotube thin films soaked at (**a**) 20 °C, (**b**) 40 °C, (**c**) 50 °C, and (**d**) 60 °C in 5 mM CdSe bath solution for 1 h (A = anatase TiO_2_, T = Ti metal).

**Figure 4 materials-13-02533-f004:**
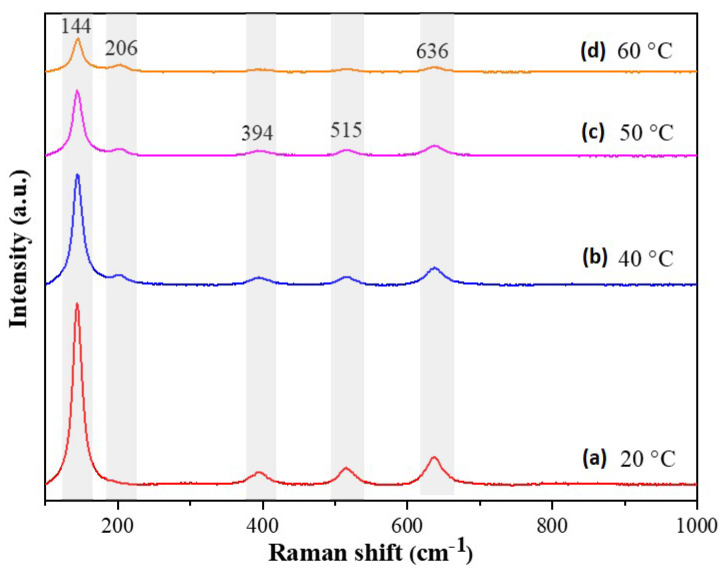
Raman spectrum of CdSe–TiO_2_ nanotube thin films soaked at (**a**) 20 °C, (**b**) 40 °C, (**c**) 50 °C, and (**d**) 60 °C in 5 mM CdSe solution for 1 h.

**Figure 5 materials-13-02533-f005:**
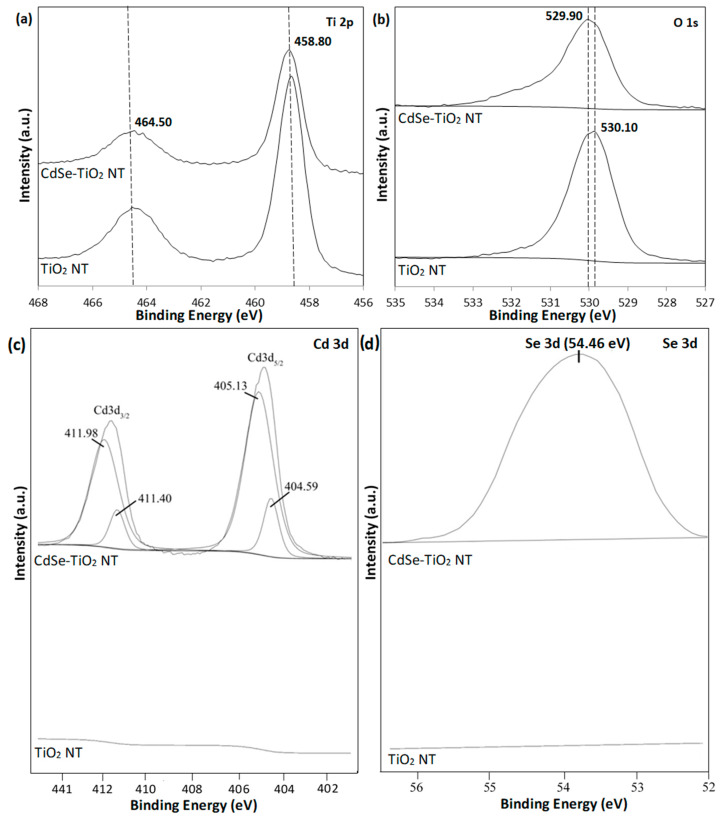
High-resolution XPS spectra of (**a**) Ti 2p, (**b**) O 1s, (**c**) Cd 3d, and (**d**) Se 3d of pure TiO_2_ nanotube thin films and CdSe–TiO_2_ nanotube thin films.

**Figure 6 materials-13-02533-f006:**
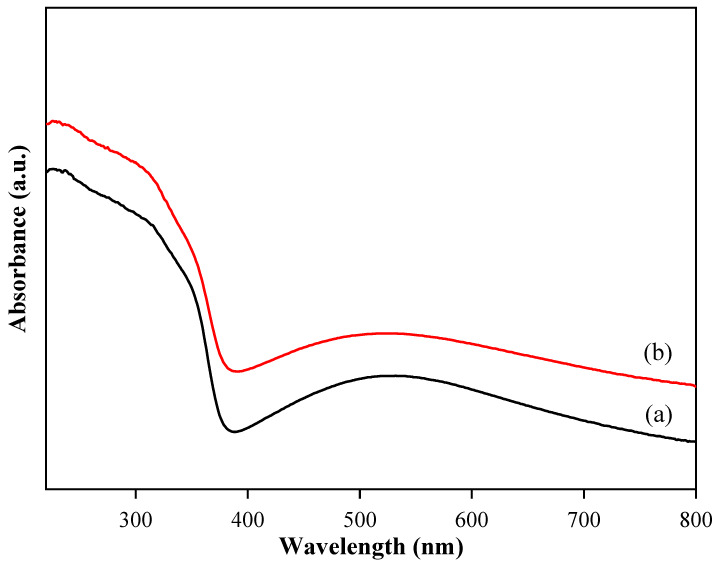
Absorption patterns for (**a**) CdSe–TiO_2_ nanotube thin films subjected to 20 °C soaking temperature and (**b**) pure TiO_2_ nanotube thin films.

**Figure 7 materials-13-02533-f007:**
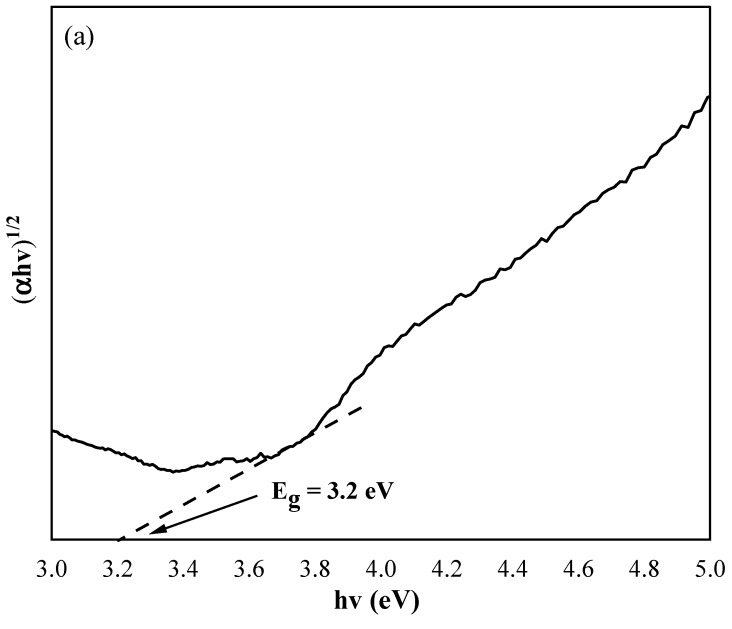
Plot of *(αhν)^1/2^* versus *hν* (eV) of (**a**) pure anatase TiO_2_ nanotube thin films and (**b**) CdSe–TiO_2_ nanotube thin films.

**Figure 8 materials-13-02533-f008:**
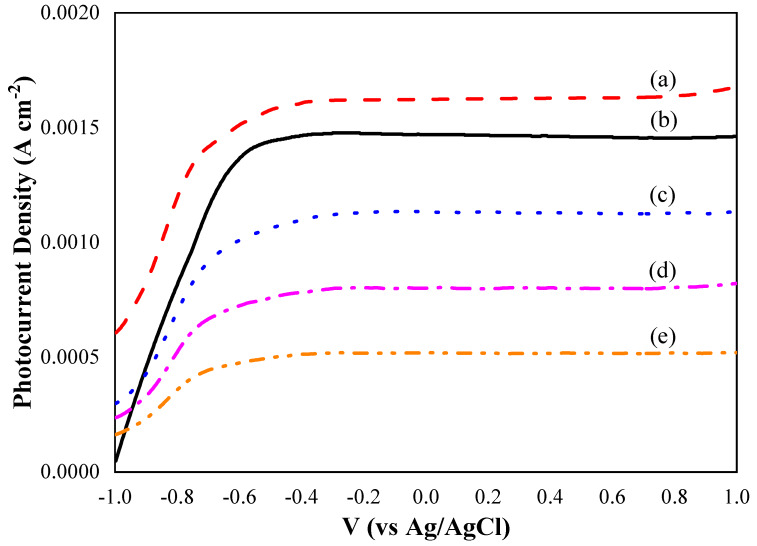
The *j_p_-V* characteristic curves of CdSe–TiO_2_ nanotube thin films soaked at different temperatures of CdSe precursor: (**a**) 20 °C, (**b**) pure TiO_2_ nanotubes thin film, (**c**) 40 °C, (**d**) 50 °C, and (**e**) 60 °C.

**Figure 9 materials-13-02533-f009:**
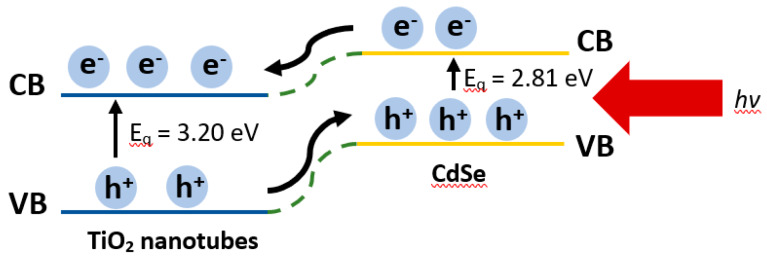
Schematic diagram representing the carrier transfer processes between CdSe and TiO_2_ nanotube thin films upon the solar light illumination. (CB and VB refer to the energy levels of the conduction and valence bands for the CdSe and TiO_2_, respectively.)

**Table 1 materials-13-02533-t001:** An average at % of CdSe–TiO_2_ nanotube thin films deposited at different chemical bath temperatures, obtained by energy dispersive X-ray spectroscopy (EDX) analysis.

Temperature (°C)	Atomic Percentage (at %)			
	Ti	O	Cd	Se
20	40.77	58.45	0.41	0.37
40	39.62	56.66	1.90	1.82
50	38.16	55.84	3.03	2.97
60	36.27	51.83	6.04	5.86

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
