# Peer review of "Influence of Temperature Reaction for the CdSe–TiO2 Nanotube Thin Film Formation via Chemical Bath Deposition in Improving the Photoelectrochemical Activity"

_materials, 2020, doi:10.3390/ma13112533_

Round 1

Reviewer 1 Report

The study by Lai et al. is a continuation of the previous report (Lai, Lau and Chou, Chem. Phys. Lett. 2019, 714, 6-10). While the published study investigated influence of soaking time on the properties and photoelectrochemical activity of CdSe-TiO2 nanotubes, the current manuscript explores temperature dependence. The manuscript is clear, the data justify the conclusion and I have only minor remarks.

In the Abstract and in the Conclusion section I have not found any information explaining why photocurrent density decreases with the increase of chemical bath temperature. Some possible explanations can be found in the manuscript text: (1) low temperature corresponds to low deposition rate and slower nuclei growth on the TiO2 nanotubes thin film (line 134) and (2) CdSe-TiO2 nanotubes thin film deposited at lower temperature shows clear surface morphology without any clogged nanotubes, hence the sample has better ion diffusion and, in consequence, enhanced photocurrent density (lines 254-257). These are very important statements, which should be emphasized better in the manuscript.

Another issue, that should be explained in more detail, is how the bath was thermostatted during the deposition process. I am also wondering if CdSe was really present in the TiO2 lattice structure, as stated in line 190, or only deposited on the TiO2 thin film.

There are also quite numerous typographic errors and grammar flubs throughout the text. Please correct them and revise the whole text carefully again.

Line 28 "nanotubes thin film thin film" --> "nanotubes thin film"

Lines 37-38 "as most promising semiconductor materials" --> "as one of the most promising semiconductor materials"

Line 42 "method to synthesis" --> "method to synthesize"

Line 54 "in order to tuning" --> "in order to tune"

Line 61 "will improved" --> "will improve"

Line 62 "reduced" --> "reduce"

Line 66 "as its use" --> "as it uses"

Line 78 "seleniosulfide" --> "selenosulfate"

Line 78 "dehydrate" --> "dihydrate"

Line 80 "the active precursor, which leads" --> "the active precursor, leads"

Line 81 "At last but not least" --> "Last but not least"

Line 86-87 "foils was performed" --> "foils performed"

Line 89 "hydrogen peroxides" --> "hydrogen peroxide"

Line 95 "sulphite" --> "sulfite"

Line 95 "saline metal powder" --> "selenium metal powder"

Line 100 "were soak vertically" --> "were soaked vertically"

Line 108 "was study" --> "was studied"

Line 145 "were taken place" --> "has taken place"

Line 157 "is appeared" --> "is collected"

Line 158 "orientation preferred orientation" --> "preferred orientation"

Line 160 "was increase" --> "increases"

Line 167 "correspond respectively with plane" --> "corresponding respectively to planes"

Line 177 "banding vibration" --> "bending vibration"

Line 180 "correspond to the first" --> "corresponding to the first"

Line 187 "Two board peaks" --> "Two broad peaks"

Line 188 "correspond to" --> "corresponding to"

Line 189 "was gradually decrease" --> "was gradually decreasing"

Line 192 "shows a slight shift" --> "show a slight shift"

Lines 204-205 "were having weak shifting" --> "had weak shifts"

Line 260 "kay redox potential" --> "key redox potential"

Line 294 "the metastable of the precursor" --> "the metastability of the precursor"

Line 297 "reduced the metastable" --> "reduced the metastability"

Line 309 "prohibited" --> "prohibit"

Line 318 "CdSe-TiO2" --> "2" should be in subscript: "CdSe-TiO2"

Line 319 "photoresponce" --> "photoresponse"

Finally, the chemical equations (1), (2), (3) and (6) should be corrected. (1), (2) and (6) were inaccurately copied from previous paper (corresponding to equations 1, 2 and 9 in Lai, Lau and Chou, Chem. Phys. Lett. 2019, 714, 6-10), while equation (3) is unfortunately erroneous also in the published article. It should be corrected as:

Cd(NH3)42+ <--> Cd2+ + 4NH3

Author Response

Reviewer 1

  1. In the Abstract and in the Conclusion section I have not found any information explaining why photocurrent density decreases with the increase of chemical bath temperature. Some possible explanations can be found in the manuscript text: (1) low temperature corresponds to low deposition rate and slower nuclei growth on the TiO2 nanotubes thin film (line 134) and (2) CdSe-TiO2 nanotubes thin film deposited at lower temperature shows clear surface morphology without any clogged nanotubes, hence the sample has better ion diffusion and, in consequence, enhanced photocurrent density (lines 254-257). These are very important statements, which should be emphasized better in the manuscript.

Sentence emphasising the surface morphology of CdSe-TiO2 nanotubes deposited at 20 oC and how the morphology affects the ion diffusion of the sample have been added in the abstract at line 29 till 31.

  1. Another issue, that should be explained in more detail, is how the bath was thermostatted during the deposition process. I am also wondering if CdSe was really present in the TiO2 lattice structure, as stated in line 190, or only deposited on the TiO2 thin film.

In this study, we use hotplate with digital programmable hotplate from Torrey Pines Scientific. The temperature was set and upon the deposition process the solution’s temperature was double check with thermometer for confirmation. The statement regarding on the temperature of the chemical bath has been added in the methodology part (CdSe-TiO2 nanotubes thin film preparation) from line 104-107.

Existence of CdSe in the TiO2 lattice structure can be observed in the XPS results. Results in Figure 5 (c) and (d) shows presence of Cd and Se in the form of Cd2+ and Se2-.. FESEM image of CdSe-TiO2 nanotubes deposited at 20 oC shows no deposition of CdSe on the surface of the nanotubes, however the XPS results prove the presence of the CdSe which believed to be in the lattice structure of the TiO2.

  1. There are also quite numerous typographic errors and grammar flubs throughout the text. Please correct them and revise the whole text carefully again.

Line 28 "nanotubes thin film thin film" --> "nanotubes thin film"

Lines 37-38 "as most promising semiconductor materials" --> "as one of the most promising semiconductor materials"

Line 42 "method to synthesis" --> "method to synthesize"

Line 54 "in order to tuning" --> "in order to tune"

Line 61 "will improved" --> "will improve"

Line 62 "reduced" --> "reduce"

Line 66 "as its use" --> "as it uses"

Line 78 "seleniosulfide" --> "selenosulfate"

Line 78 "dehydrate" --> "dihydrate"

Line 80 "the active precursor, which leads" --> "the active precursor, leads"

Line 81 "At last but not least" --> "Last but not least"

Line 86-87 "foils was performed" --> "foils performed"

Line 89 "hydrogen peroxides" --> "hydrogen peroxide"

Line 95 "sulphite" --> "sulfite"

Line 95 "saline metal powder" --> "selenium metal powder"

Line 100 "were soak vertically" --> "were soaked vertically"

Line 108 "was study" --> "was studied"

Line 145 "were taken place" --> "has taken place"

Line 157 "is appeared" --> "is collected"

Line 158 "orientation preferred orientation" --> "preferred orientation"

Line 160 "was increase" --> "increases"

Line 167 "correspond respectively with plane" --> "corresponding respectively to planes"

Line 177 "banding vibration" --> "bending vibration"

Line 180 "correspond to the first" --> "corresponding to the first"

Line 187 "Two board peaks" --> "Two broad peaks"

Line 188 "correspond to" --> "corresponding to"

Line 189 "was gradually decrease" --> "was gradually decreasing"

Line 192 "shows a slight shift" --> "show a slight shift"

Lines 204-205 "were having weak shifting" --> "had weak shifts"

Line 260 "kay redox potential" --> "key redox potential"

Line 294 "the metastable of the precursor" --> "the metastability of the precursor"

Line 297 "reduced the metastable" --> "reduced the metastability"

Line 309 "prohibited" --> "prohibit"

Line 318 "CdSe-TiO2" --> "2" should be in subscript: "CdSe-TiO2"

Line 319 "photoresponce" --> "photoresponse"

All suggested grammatically errors have been corrected.

  1. Finally, the chemical equations (1), (2), (3) and (6) should be corrected. (1), (2) and (6) were inaccurately copied from previous paper (corresponding to equations 1, 2 and 9 in Lai, Lau and Chou, Chem. Phys. Lett. 2019, 714, 6-10), while equation (3) is unfortunately erroneous also in the published article. It should be corrected as:

Cd(NH3)42+ <--> Cd2+ + 4NH3

The chemical equations have been corrected

Reviewer 2 Report

In this manuscript, a low-temperature chemical bath deposition technique was used for the deposition of CdSe particles on the TiO2 nanotubes thin film and the photoelectrochemical performance of the sample was tested. Interestingly, a significant enhancement in photocurrent density was observed for CdSe-TiO2 nanotubes thin films than pure TiO2 nanotubes thin films. This work contains some interesting results, but it needs some modifications in the present state. Therefore, a major revision should be made before its publication.

  1. The introduction establishes the context of the work, but in the present state, it does not provide sufficient justification for this study. It should be rewritten to expound the research significance of the present work.
  2. The authors should provide a schematic representation of the formation mechanism of CdSe-TiO2 nanotubes thin films.
  3. On what basis the reaction pH was adjusted between 12 to 12.5. The pH of the solution normally varies from precursor to precursor. The authors must justify the selection of pH, temperature and time.
  4. “The agglomeration of CdSe into larger particles was observed when the chemical bath temperature increases to 50 and 60°C”. What is the significant reason for this formation?
  5. In the EDX measurement results, atomic percentages of Cd and Se are presented in the range of 0.37 to 0.41 atomic percentages (Fig. 4). What is the error limit for Cd and Se detection in EDX measurement?
  6. The authors are advised to add line spectra in Fig. 2 (XRD) by representing the both CdSe and TiO2 phases with JCPDS numbers.
  7. What is the relationship between the optical band gap and PEC activity under light irradiation?
  8. What is the key factor (e.g., surface area, chemical composition, morphology) affecting the PEC performance?
  9. Some evidences like EIS, and IT results should be provided to confirm the photocurrent density enhancement.
  10. The explanation of the PEC mechanism needs more clarification. The authors are advised to explain the mechanism with neat schematic representation.
  11. In the current state, there are more typographical errors and the language should be improved. Therefore, the authors are advised to recheck the whole manuscript for improving the language and structure carefully.

Author Response

Reviewer 2

  1. The introduction establishes the context of the work, but in the present state, it does not provide sufficient justification for this study. It should be rewritten to expound the research significance of the present work.

The research significant of the present work has been included at the end part of the introduction.

  1. The authors should provide a schematic representation of the formation mechanism of CdSe-TiO2 nanotubes thin films.

The formation mechanism of CdSe-TiO2 nanotubes thin films has been discussed in details in the discussion part.  We have added schematic diagram of the preparation of TiO2 nanotubes thin film and CdSe- TiO2 nanotubes thin film in the manuscript

  1. On what basis the reaction pH was adjusted between 12 to 12.5. The pH of the solution normally varies from precursor to precursor. The authors must justify the selection of pH, temperature and time.

pH of CdSe solution was 5.7 while concentrated ammonia solution (30 %) was at pH of 12. pH of the CdSe solution was adjusted between 12 to 12.5 during the addition of concentrated ammonia solution to prevent reverse reaction of Cd(NH3)42+ to form stable cadmium hydroxide (Cd(OH)2) at higher pH. Therefore, it is crucial to adjust and maintain the pH between 12 to 12.5.

  1. “The agglomeration of CdSe into larger particles was observed when the chemical bath temperature increases to 50 and 60°C”. What is the significant reason for this formation?

As the chemical bath temperature increase, the reaction rate will increase. Therefore, an enhancement of amount of CdSe deposited onto TiO2 nanotubes was observed. As the deposition process proceed at higher reaction rate, the deposition of CdSe was occurred at the previously deposited CdSe nuclei leading to the growth of CdSe into larger agglomerate.

  1. In the EDX measurement results, atomic percentages of Cd and Se are presented in the range of 0.37 to 0.41 atomic percentages (Fig. 4). What is the error limit for Cd and Se detection in EDX measurement?

Error limit for EDX measurement is 1 ppm or 0.1 wt. %. In this study, the wt. % of Cd and Se for CdSe-TiO2 nanotubes deposited at temperature 20 oC are 1.56 wt. % and 0.98 wt. % respectively, which is above the detection limits of EDX.

  1. The authors are advised to add line spectra in Fig. 2 (XRD) by representing the both CdSe and TiO2 phases with JCPDS numbers.

Line spectra have been added in the Fig. 2 with their plane as shown in figure below. JCDPS numbers have been provided in the discussion part.

Figure 2. XRD diffraction patterns of CdSe-TiO2 nanotubes thin film soaked at (a) 20 °C, (b) 40 °C, (c) 50 °C and (d) 60 °C in 5 mM CdSe bath solution for 1 h [A = anatase TiO2, T = Ti metal].

  1. What is the relationship between the optical band gap and PEC activity under light irradiation?

Deposition of CdSe heterostructures to TiO2 nanotubes will increase of the absorption in the visible light region. This implied that the absorption of the TiO2 nanotubes have been extended into the visible light region through the deposition of CdSe. As such, CdSe- TiO2 nanotubes will absorb more visible light in order to generate more electron-hole pairs, hence increase the photocurrent response of the sample. This explanation has been written in the manuscript at line 222 to 225.

  1. What is the key factor (e.g., surface area, chemical composition, morphology) affecting the PEC performance?

The key factor affecting the PEC performance is the morphology and surface area of the sample. In this study, TiO2 nanotubes was used as the main material which provides a high surface area for the ion diffusion and electron transfer of the sample. However, due to the large band gap possessed by TiO2 incorporation of CdSe was done to alter the band gap of the sample. The morphology of deposited CdSe plays a huge role in determining the surface area of the composite material (CdSe-TiO2 nanotubes thin film). As the chemical bath temperature increase, the CdSe tends to become agglomerates and covered most of the nanotubes opening. Therefore, the surface will decrease and resulting poor photoecurrent density. Therefore, surface area and the morphology are the key factor affecting the PEC performance.

  1. Some evidences like EIS, and IT results should be provided to confirm the photocurrent density enhancement.

EIS result will give the resistive and capacitive properties of the material. It also can generate the equivalent circuit model for the sample. However, we believed that this analysis is not crucial for this study. Photocurrent density can be calculated based on the PEC results.

  1. The explanation of the PEC mechanism needs more clarification. The authors are advised to explain the mechanism with neat schematic representation.

The schematic representation in Figure 9 have been replaced with much clearer figure and the explanation of the PEC mechanisms have been rewrite with more clarification from line 317 to 334.

  1. In the current state, there are more typographical errors and the language should be improved. Therefore, the authors are advised to recheck the whole manuscript for improving the language and structure carefully.

We have rechecked the whole manuscript upon resubmission. However, we could not submit this manuscript for proofreading service as me have limited time to do revision.

Reviewer 3 Report

I generally do not have any concern about the manuscript. I believe manuscript is written well. I have suggestion for authors, If authors wanted to include flowchart for the preparation of tubes in the procedure. They can do some work on that. It is totally on authors. 

Author Response

Reviewer 3

  1. I generally do not have any concern about the manuscript. I believe manuscript is written well. I have suggestion for authors, If authors wanted to include flowchart for the preparation of tubes in the procedure. They can do some work on that. It is totally on authors. 

Schematic diagram of the preparation of TiO2 nanotubes thin film and CdSe- TiO2 nanotubes thin film have been added in the manuscript.

Figure 1. Schematic diagram of the preparation of TiO2 nanotubes thin film and CdSe- TiO2 nanotubes thin film

Reviewer 4 Report

Please allow me to review  this very great valuable, interesting and important manuscript which is very suitable to “Materials” reporting important research area associated with the CdSe-TiO2 nanotube thin film as a promising material for photoelectrochemical cell application. .

I am for that this work described by the authors has confirmed that small amount of CdSe is enough to enhance photoelectrochemical performance of the sample.

I feel that the contents in this paper are certainly novel and the quality of the work is relatively high. The technical validity of the characterization work is relatively good. I feel that the authors are expert in these related fields.  I can confirm that the characterizations appear to be very well carried out. I am very happy to recommend publication as original paper in the “Materials”.  The authors may wish to take into account the following points when putting together a final, publishable, version of the manuscript.

1) As to Table 1, would the authors please describe in details how to get these atomic percentage? How about errors of the values?

2) I feel that references should be more up-dated. I feel that references might be unfair since many research groups have been developing various related materials on various aspects of material sciences. Are there suitable references published in “Materials”?

3) Would the authors please describe in details how and what relationship between the results of Figure 1 about FESEM images and the other Figures have?

4) Would the authors please describe in details what and how important “CBD method” is in this paper?

5) Are there any other very powerful tools for the nanotubes thin file in this paper?

Figure 3 is for Raman data.  How about IR?

6) I feel that Figure 8 might be high light points in this paper. Would the authors describe in more details about Figure 8?  What data did support Figure 8?

7) Are there any other supplemental data for this very valuable paper?

8) How did the authors chemical bath temperature precisely?

9) Line 156-161, would the authors please describe in more details to be understood easily?

10) Line 281-289, are there any directly evidences for these reactions?

I just have found that (3) in line 283 and (6) in line 289 are not right and should be corrected.
Cd(NH3)2 should be Cd(NH3)4

Author Response

Reviewer 4

  1. As to Table 1, would the authors please describe in details how to get these atomic percentage? How about errors of the values?

The atomic percentage of this samples were obtained from the EDX analysis itself. However, the atomic percentage can be calculated from weight percentage using formula below:

Error limit for EDX measurement is 1 ppm or 0.1 wt. %. In this study, the wt. % of Cd and Se for CdSe-TiO2 nanotubes deposited at temperature 20 oC are 1.56 wt. % and 0.98 wt. % respectively, which is above the detection limits of EDX.

  1. I feel that references should be more up-dated. I feel that references might be unfair since many research groups have been developing various related materials on various aspects of material sciences. Are there suitable references published in “Materials”?

The references have been updated

  1. Would the authors please describe in details how and what relationship between the results of Figure 1 about FESEM images and the other Figures have?

Additional discussion on the relationship between the result of FESEM images and the other figures have been added in the discussion part of the manuscript from line 174 to 175.

  1. Would the authors please describe in details what and how important “CBD method” is in this paper?

CBD method is one of a deposition technique to deposit metal chalcogenide thin films. This study used TiO2 nanotubes as the main substrate to deposit CdSe. The main objective of this study is to deposit CdSe onto the TiO2 nanotubes in order to enhance the photoelectron properties of the sample but at the same time to maintain the high surface area of the TiO2 nanotubes. This manuscript give an insight of influence of chemical bath temperature on the physical, chemical and electrochemical properties of the sample. As per our knowledge, to date, there is no report on the influence of chemical bath temperature for the formation of CdSe- TiO2 nanotubes thin film.

  1. Are there any other very powerful tools for the nanotubes thin file in this paper?

Figure 3 is for Raman data.  How about IR?

Raman and XPS is a very powerful tools to investigate the chemical bonding and chemical states for this samples. However, we could not do the analysis as we do not have enough resources and financial. IR analysis is analysis technique to analysed the functional group of a solid (particles), liquid and gas sample. However, our samples are in a form of thin film and it was not advisable to IR analysis as the result will be not accurate.  

  1. I feel that Figure 8 might be high light points in this paper. Would the authors describe in more details about Figure 8?  What data did support Figure 8?

The explanation of the PEC mechanisms has been rewritten with more clarification from line 317 to 334.

  1. Are there any other supplemental data for this very valuable paper?

There is no supplementary data for this manuscript.

  1. How did the authors chemical bath temperature precisely?

In this study, we use hotplate with digital programmable hotplate from Torrey Pines Scientific. The temperature was set and upon the deposition process the solution’s temperature was double check with thermometer for confirmation. The statement regarding on the temperature of the chemical bath has been added in the methodology part (CdSe-TiO2 nanotubes thin film preparation) from line 104-107.

  1. Line 156-161, would the authors please describe in more details to be understood easily?

Line 156-161 have been rewritten

  1. Line 281-289, are there any directly evidences for these reactions?

The reaction has been proposed based on the CDB reaction mechanisms reported by Pawar et al. In addition, this reaction mechanisms was published in our previous report which have been cited in the manuscript.

  1. (3) in line 283 and (6) in line 289 are not right and should be corrected. Cd(NH3)2 should be Cd(NH3)4

The equation has been corrected

Round 2

Reviewer 2 Report

The manuscript is improved in the revised version and it can be accepted in the present form.
